# Knowledge, risk of infection, and vaccination status of hepatitis B virus among rural high school students in Nanumba North and South Districts of Ghana

Awolu Adam[1,2]*, Adam Fusheini[2,3]

**1** School of Public Health, University of Health and Allied Sciences, Ho, Ghana, **2** Center for Health Literacy and Rural Health Promotion, Accra, Ghana, **3** Department of Preventive and Social Medicine, Dunedin School of Medicine, University of Otago, Dunedin, New Zealand

* aawolu@uhas.edu.gh

## Abstract

### Background

Hepatitis B (HB) is a viral infection that affects the liver and can lead to life-threatening conditions including cirrhosis and liver cancer. Over a billion people are estimated to be infected globally with the hepatitis B virus, with over 240 million chronically infected. Sub-Saharan Africa including Ghana is an HBV endemic area and an estimated 5%– 10% of the population in the region is infected. Research on the knowledge and vaccination status of hepatitis B in rural communities in Ghana is lacking.

### Objectives

The objectives of this study proposed were to assess the HBV knowledge, risk of HBV infection, and vaccination status of high school students in two rural districts of the Northern region on Ghana.

### Methods

A cross-sectional study of a random sample of 426 students from two senior high schools in the Nanumba North and South districts of the Northern region of Ghana on hepatitis B knowledge and vaccination status was conducted. Descriptive statistics were used to analyze and present data on demographic and knowledge variables. A Mann Whitney U test was used to compare the differences in HBV knowledge between male and female students and between students of the two high schools that were involved in the study. Pearson correlation coefficient was used to compute the association between HBV knowledge and age of students. Logistic regression was used to develop a model to predict variables that influenced vaccination against HB.

**Data Availability Statement:** Limited dataset for the study has been deposited in Zenodo and the URL is https://zenodo.org/record/3594143#.XgZ0TZNKjIU.

**Funding:** The authors received no funding for this work

**Competing interests:** The authors have declared that no competing interest exist.

## Results

The results of the study showed basic but not a good knowledge of HBV among the rural high school students, with a mean score of 11.8 (SD = 1.98) out of a maximum score of 16. Descriptive statistics also revealed that only 20% of 426 students ever tested for HBV and 96 (22.5%) were vaccinated against HBV. A Mann-Whitney U test results revealed no statistically significant difference in HBV knowledge between male and female students ($p = 0.688$, two-tailed) and between the two high schools ($p = 0.24$, two-tailed). A Pearson correlation showed no relationship between age and HBV knowledge ($p = 0.486$). Regression analysis showed that only taking the HBV test ($p < 0.05$) and attending Bimbilla Senior High ($p = 0.032$) significantly predicted vaccination against HBV infection.

## Conclusion

The results of this study has re-echoed the high prevalence of HBV in Ghana. The poor state of knowledge and a high risk of HBV infection among young adults in rural communities have also been highlighted in the findings of this study. Vaccination against the HBV infection was found to be low and consistent with other findings. Finally, HBV screening is shown to be significantly associated with vaccination against the virus, hence the need for national screening and vaccination programs.

## Introduction

Viral hepatitis B (HBV) infection is a serious health condition and constitutes major public health challenges globally, especially in developing countries. The Center for Disease Control and Prevention (CDC) described hepatitis B as a liver infection caused by HBV that can be transmitted through blood, semen, and other bodily fluids, and from a mother to her newborn through birth [1]. Hepatitis B infection is potentially life-threatening as it can lead to chronic and debilitating health conditions including cirrhosis and liver cancer, and also death [1,2]. The risk of developing chronic hepatitis B infection is dependent on the age at which an individual is infected. For instance, the CDC estimated that 90% of infected infants will become chronically ill compared to 2%– 6% of those who are infected as adults [1].

There is consistent evidence of increasing prevalence of HBV infection globally with an estimate of about one-third of the world's population being infected with hepatitis B virus (HBV) and around 290 million being chronic carriers [3, 4] despite the availability of effective vaccines against the virus in most countries [4, 5]. Yet, unlike HIV/AIDS, HBV has not received similar attention, especially in the endemic regions of sub-Saharan Africa and Asia. There are differences in the global estimates of chronic HBV prevalence but all point to a serious global problem, particularly in sub-Saharan Africa and East Asia. For instance, the World Health Organization (WHO) estimated that in 2015, 257 million people were living with chronic hepatitis B infection, which is defined as hepatitis B surface antigen-positive, and the estimated deaths due to HBV infections were 887,000 globally [2]. Most of the HBV-related deaths were from cirrhosis and hepatocellular carcinoma. Other estimates indicate that 2 billion people globally have been infected with HBV, with more than 350 million developing chronic and lifelong infections [3]. HBV and hepatitis C are root causes of liver cancer, leading to 1.34 million deaths every year [2]. In fact, all estimates of HBV prevalence show that HBV is more

infectious than even HIV, and those affected by HBV far outnumber those afflicted with HIV/AIDS since its peak in the 1980s [6].

Like HIV/AIDS, HBV is endemic in sub-Saharan Africa and East Asia where societies are poverty afflicted and the knowledge about the disease is lowest, including Ghana. According to WHO, HBV prevalence is highest in the WHO Western Pacific Region and the WHO African Region, where 6.2% and 6.1% of the adult population is infected, respectively [2]. Ghana is equally an HBV endemic country as a number of screening programs reported a high prevalence rate. For instance, researchers used the Wondfo HBsAg test kit to test the prevalence of HBsAg among blood donors at the Tamale Teaching Hospital in Ghana and reported that those who donated blood were two groups—voluntary donors (576) and blood replacement donors (5,878)—making up a total of 6,462 [4]. The researchers reported that 10.79% of the voluntary donors and 11.59% of the replacement donors were HBsAg positive [4]. This was just one institution and the rate is indicative of a bigger problem as it relates to only health facility reports. Another screening program test among 1,500 pregnant women in the Eastern region of Ghana reported 10.6% hepatitis B prevalence among the participants with some variations in the districts covered [5]. Again, among 838 HIV positive persons in Ghana, researchers used rapid test kits to test the prevalence of co-infection of HBV and HIV and reported an overall prevalence of 16.7% [7]. More recently in 2019, researchers have reported 8.5% and 14.2% prevalence of chronic HBV and occult hepatitis, respectively isolated from the 305 dried blood spots collected in Ghana [8]. The above prevalence rates corroborate WHO's report of 5%– 10% adult prevalence rates in sub-Saharan Africa.

Analysis of the HBV prevalence/incidence data from the Ghana Health Service (GHS) district health information management systems (DHIMS2) shows a worrying trend. Organized or standardized data on hepatitis B was lacking until the implementation of DHIMS by GHS in 2010. The DHIMS data on reported HBV infection is categorized into suspected cases, confirmed cases, and death resulting from HBV complications. The trend shows an increasing incidence/prevalence in the country. For example, the confirmed cases in 2012, 2013, and 2014 nationally as reported were 3508, 4419, and 7324, respectively [9]. Reported HBV-related deaths for the three years were 164, 101, and 131, respectively [9]. It is worth noting that the suspected cases were far more than the confirmed cases in these years, and also that the figures reported here were institution-based reported cases only. There are also regional variations in both the suspected and confirmed cases of HBV prevalence as well as HBV-related deaths. Using the confirmed HBV infection cases in 2014 for instance, Ashanti, Brong Ahafo, Greater Accra, Northern, and Upper West regions had 918, 1200, 2155, 288, and 1124 cases, respectively. It could, therefore, be inferred that HBV is endemic in Ghana.

The transmission of HBV from one person to another can be through unprotected sex, blood transfusion, mother-to-child transmission, and through other bodily fluids. Unprotected sex, blood transfusion, and mother-to-child transmission have been reported to be the major routes of HBV infection in Ghana [4]. Unsafe drug use, especially intravenous drug use, has been linked to hepatitis B infection and so those who engaged in unsafe drug use have an increased risk of hepatitis B infection. HBV, therefore, seems to have more transmission routes than many sexually-transmitted diseases including HIV. However, mother-to-child transmission appears to be one of the greatest risks of HBV infection in children. Hepatitis B virus (HBV) infection in a pregnant woman poses a serious risk to her infant at birth. This is because without postexposure immunoprophylaxis approximately 40% of infants born to HBV-infected mothers in the United States will develop chronic HBV infection, approximately one-fourth of whom will eventually die from chronic liver disease [10]. By the age of 6 months, there is the risk of chronic HBV infection in 90% of the infants whose mothers have the hepatitis B surface antigen (HBsAg) and hepatitis B e antigen if no immunoprophylaxis is done [11].

HBV has also been found as a co-infection among people living with HIV and other diseases in Ghana. In a systematic review of studies on HIV/HBV co-infection conducted across most regions in Ghana, researchers found an HIV/HBV co-infection rate of between 2.4% and 13.6% with a general co-infection rate of 13.6% [12]. In studying the etiology of viral hepatitis in 155 jaundiced patients in a teaching hospital in Ghana, researchers found that 54.2% of them tested positive for hepatitis B and 32.9% tested positive for hepatitis E [13]. The risk of infection to any disease or condition is dependent upon several factors, which may reduce or increase the risk. These correlates of factors may include behavioral, biological, and socio-cultural factors.

The most effective and efficient way to prevent HBV infection in a population is through vaccination. Unfortunately, HBV awareness, access to screening, vaccination, and treatment have remained poor in resource-limited countries due to poverty, illiteracy, and the lack of political will [14]. In 2003, HBV vaccination was introduced for all newly born in Ghana and the screening of pregnant women to prevent mother-child transmission is now ongoing. Prior to 2003, there was no national HBV vaccination and so all those born before 2003 had the greatest risk of mother-to-child transmission. However, the introduction of the immunization policy has resulted in fewer infections and lower prevalence among those who were born after the introduction of the HBV immunization policy. To evaluate the impact of the immunization policy on HBV prevalence in the Cape Coast Municipality of the Central Region of Ghana, researchers sampled 501 school children in a pilot study and found that the prevalence of HBcAb was 2.6% and 6.1% among those born after the introduction of the policy and those born prior to the introduction of the policy, respectively [15]. This is a significant difference in the infection rates of those born before the mandatory HBV vaccination policy was introduced and those born after the introduction of the policy. This is attestation that immunization of newborn babies is critical to ending the HBV epidemic in Ghana.

There is currently no national HBV screening program for the adult population and young people born before 2003. Only episodic screenings are done in the general population mainly by non-profit organizations. Educational programs are lacking in terms of HBV in Ghana coupled with inadequate attention or resource commitment. As a result of poor attention, knowledge about HBV as well as the HBV vaccine is poor in many developing countries. In examining knowledge, accessibility, and HBV vaccination status of 643 pregnant women at three levels of healthcare delivery in Ibadan, Nigeria, researchers reported that 76% of the pregnant women had very low levels of knowledge of HBV, 20% reported being screened, and only 9.7% vaccinated against HBV infection [14]. In the study, being a healthcare worker, attaining higher education, and seeking healthcare at tertiary levels increased the likelihood of improved knowledge, previous screening, and vaccination against HBV infection. Among 370 health workers sampled in the Bahir City in Ethiopia, only 52% scored above average knowledge about HBV infection, 62% exhibited adequate knowledge about HBV vaccine, and only 5.4% completed three doses of HBV vaccination [16]. In a study of knowledge, perceptions, and practices about HBV among the general population of France, both HBV knowledge and vaccination were found to be poor [16]. Using a telephone interview, the researchers interviewed 9,014 individuals between the ages 18 and 69 years and reported that compared to HIV, knowledge of HBV was poor and only 27.4% ever screened for HBV [17]. Numerous similar findings of poor HBV knowledge and low vaccination have been reported in other settings [18, 19, 20, 21]. Therefore, the poor knowledge and low vaccination rates seem pervasive globally especially in developing countries.

A few studies on HBV in Ghana have been analyzed but almost all the studies tested prevalence rates among some selected groups in the country or conducted short screenings. Only a few of the studies examined knowledge, attitudes, and practices in relation to HBV. For

**Table 1. Hepatitis B Knowledge questionnaire's responses in relative percentages.**

| Question | Frequency | Relative Percent |
|---|---|---|
| 1. Hepatitis B is a disease caused by: | | |
| Virus | 295 | 69.2 |
| Bacteria | 102 | 23.9 |
| Fungi | 29 | 6.8 |
| 2. Mosquito bites can give people Hepatitis B. | | |
| True | 199 | 46.7 |
| False | 227 | 53.3 |
| 3. People can get Hepatitis B through physical contact such as shaking hands and hugging. | | |
| True | 236 | 55.4 |
| False | 190 | 44.6 |
| 4. Having sexual intercourse with someone who has hepatitis B without condoms can give you hepatitis B. | | |
| True | 334 | 78.4 |
| False | 92 | 21.6 |
| 5. Having unprotected sex with many people increases the risk of getting infected with the hepatitis B virus. | | |
| True | 349 | 81.9 |
| False | 77 | 18.1 |
| 6. Someone can get hepatitis B through receiving infected blood in his or her body at the hospital. | | |
| True | 382 | 89.7 |
| False | 44 | 10.3 |
| 7. Someone can get hepatitis B by sharing needles, syringes, and razors with infected persons. | | |
| True | 357 | 83.8 |
| False | 69 | 16.2 |
| 8. Can a person give hepatitis B to other people but not know he/she has hepatitis B? | | |
| Yes | 357 | 83.8 |
| No | 69 | 16.2 |
| 9. Hepatitis B can spread through food. | | |
| True | 228 | 53.5 |
| False | 198 | 46.5 |
| 10. Pregnant women can pass on the hepatitis B virus to their newborn babies through birth. | | |
| True | 328 | 77 |
| False | 98 | 23 |
| 11. Children can get infected with the hepatitis B virus through breastfeeding. | | |
| True | 325 | 76.3 |
| False | 101 | 23.7 |
| 12. The best way to find out if you have hepatitis B is through: | | |
| Blood test | 354 | 83.1 |
| Urine test | 35 | 8.2 |
| Physical appearance | 37 | 8.7 |
| 13. Hepatitis B is a serious disease that can lead to liver damage, liver failure, liver cancer, or even death. | | |
| True | 396 | 93.0 |
| False | 30 | 7.0 |

(*Continued*)

**Table 1.** (Continued)

| Question | Frequency | Relative Percent |
|---|---|---|
| 14. The best way to prevent hepatitis B is by: | | |
| Getting vaccinated | 344 | 80.8 |
| Abstaining from sex | 74 | 17.4 |
| Taking your medication | 8 | 1.9 |
| 15. There is a vaccine that can protect people against the hepatitis B virus. | | |
| True | 382 | 89.7 |
| False | 44 | 10.3 |
| 16. How many injections do you need to be protected against Hepatitis B? | | |
| Correct answer (3) | 127 | 29.8 |
| Wrong answer | 297 | 69.7 |

instance, in a cross-sectional study of HBV knowledge, attitudes, and practices (KAP) among 175 healthcare workers at the Suntreso Government Hospital, Ghana, researchers reported basic knowledge or awareness of HBV with age, occupation, and years of experience being the significant predictors of KAP. Using qualitative methods, researchers examined the perceptions and understanding of HBV in the Upper West region of Ghana and found extremely low levels of knowledge and pervasive misconceptions about HBV [22]. In addition to this, the researchers also reported that the participants had no access to HBV immunization, testing, and vaccination services. Among barbers in Obuasi municipality, a high prevalence (14.5%) of HBV infection was reported [23]. The researchers also reported that 90.5% had heard of HBV but lacked knowledge of transmission routes and prevention methods [23]. A similar study among pregnant women attending antenatal care in a mission hospital in Ghana reported 10.2% HBV prevalence, high awareness, and poor accurate knowledge about HBV [24].

However, no study on HBV knowledge, vaccination status, perceptions, or attitudes was found to have been conducted in the Northern region and more so in the Nanumba North and Nanumba South districts of the region. No evidence of such a study among young adults in and out of school in the region and the districts are located in the literature. In order to design and implement interventions to prevent the spread of infections and to promote voluntary testing and vaccination against HBV, the gap in knowledge must first be bridged. This study was, therefore, designed to assess the rural high school students' knowledge, perceptions, and vaccination status of HBV in the Nanumba North and South districts.

## Methods

### Design and setting

In order to achieve the objectives of the study, we employed a descriptive, quantitative, cross-sectional design, which utilized a standardized, closed-ended questionnaire with a few open-ended respondents' self-administered questionnaires to assess the knowledge, risk, and vaccination status of high school students in two rural districts of the Northern region of Ghana. In order to assess HBV knowledge, we included 16 questions in the questionnaire as knowledge variables. The 16 questions were drawn from information about HBV by CDC to develop the questionnaire and pre-tested as indicated above. A mark of one (1) was assigned to each correct answer and zero was assigned to each wrong answer. The number of correct answers a participant selected determined how much knowledge she or he possessed about the hepatitis B virus. The questionnaire captured the 16 questions on knowledge as seen in Table 1 and six risk factor items as

detailed in Table 3. The vaccination status was captured by a key question that sought to find out if the respondents ever received or at least recall being vaccinated against the hepatitis B virus. Knowledge in the study was defined respondents knowing the causal agent of the virus, the means and modes of infection and spreading of the virus, and the most effective methods of prevention. For the purpose of this study, knowledge was categorized into poor, basic, and good. A score of less than six (6) correct answers out of the 16 questions was considered as poor knowledge. A score between 6 and 12 correct answers was considered basic knowledge, and a score of 13–16 was considered good knowledge of HBV. The number of correct answers a participant selected determined how much knowledge she or he possessed about the hepatitis B virus.

As stated earlier, the study was conducted in the Nanumba North and South Districts. The two are part of the sixteen administrative districts in the region. Nanumba North has two senior high schools—one public and one private—while Nanumba South District has one senior high school. The two schools were purposively selected as they have similar population sizes and are both public schools.

## Sample size determination

The sample size was based on the population of the two high schools combined, the prevalence rate of HBV in Ghana, and the power calculation. The combined population of the two senior high schools proposed for this study was 3051 students. The sample size was derived using the sample determination formula by Kothari [25] and a web-based sample size calculator by Raosoft Inc. [26]. Both approaches were set at 95% confidence interval and yielded a similar sample size of 423 participants. A 10% non-response rate was calculated and added, which increased the estimated sample size to 465.

## Sampling method

We employed a proportionate sampling method to draw 426 (92%) students from Bimbilla Senior High (BSH) in the Nanumba North and Wulensi Senior High (WSH) in the Nanumba South districts of the Northern Region. Proportionate sampling technique is where samples are drawn to be representatives of population sub-groups and the sampling fraction in each stratum is made equal to the sampling fraction for the total population [27]. Proportionate sampling was preferred because of the involvement of two high schools from two districts. We needed to have a representation of the population sizes of the schools. Therefore, 204 students were drawn from BSH and 222 were drawn from WSH. A register (roll) of all the students was obtained from the schools' authorities, which was used as a sampling frame. Since senior high school education in Ghana is three years, the stratified sampling technique was employed in grouping the students into three stratums according to their year of study (1, 2, & 3). The selection of respondents into the study was then done on the basis of these years of study. After a proportional stratification, simple random sampling was done by writing YES and NO for the selected students to pick from a container. YES implied being recruited into the study and all the numbers corresponding to the name list on the roll were selected until the stipulated

**Table 2. Hepatitis B knowledge categories.**

|       | Knowledge           | Frequency | Percent |
|-------|---------------------|-----------|---------|
| Valid | Poor HBV knowledge  | 7         | 1.6     |
|       | Basic HBV knowledge | 258       | 60.6    |
|       | Good HBV knowledge  | 161       | 37.8    |
|       | Total               | 426       | 100     |

sample size from the respondents who met the expressed qualification criteria were met. This ensured a fair distribution for all levels to participate in the study.

## Pretesting of questionnaire

The structure developed was pretested on 25 students selected from both BHS and WSH. This number included 13 students from BHS and 12 students from WSH. Both male and female students were represented in the pretest sample. Changes were made to the original questionnaire by rewriting sentences and changing words and technical terms that the students did not understand. The students who participated in the pretest were excluded in the final sample recruited for the study.

## Data collection procedure

Following the successful selection of qualified and eligible respondents, they were then sent to different classes in the schools where the researchers again explained the purpose and procedures of the study to them. They were also told of voluntary withdrawal if they wanted to withdraw after which they were given self-administered questionnaires to complete. The questionnaires were written in the English Language at the 5th-grade reading level. Data was collected in February 2018.

## Data management and analysis

All data entry, analysis, and presentation were done using SPSS version 20 graduate pack. Descriptive statistics were used to analyze demographic variables and knowledge-based questions. A total knowledge score was computed and used to categorize participants into poor, basic, and good knowledge categories. We conducted a Mann-Whitney's U test to determine the differences between males and females and between the two high schools involved in the study in terms of HBV knowledge. The Pearson correlation test was used to test whether there was a relationship between the age of participants and HBV knowledge in the study sample. To examine the risk of hepatitis B infection among the participants in this study, six questions were asked that largely covered behavioral and biological attributes. We developed a model using logistic regression and the predictor variables we included were previous testing for the hepatitis B virus, gender, knowledge score, number of sexual partners, whether or not the participant was diagnosed with or having a sexually transmitted disease, and the high school the student came from. All statistical analysis was conducted at a 95% confidence interval.

## Ethical approval

Ethical review of the study was done and approved by the Ethics Committee of Ghana Health Service (GHSEC) with the approval number GHS-ERC: 01/05/17 in 2017 and the data was collected in February 2018. The district directorates of education of both Nanumba North and South districts, as well as the headmasters of the two high schools, gave permission before the study was conducted. Each student was provided with a written information sheet and informed consent form and all those recruited into the study had to voluntarily agree or assent to participate in the study and sign the form before completing the questionnaire. Parental consent for minors was waved because the permission granted by the principals of the high schools and the directories of education for the two districts addressed that.

## Findings

Important findings were made in this study and these findings are organized in sub-headings including demographic characteristics, knowledge of hepatitis B, risk of hepatitis B infection, and the vaccination status of participants against hepatitis B virus.

### Demographic characteristics of participants

The participants of the study were high school students sampled from two rural high schools in the Nanumba North and South districts of the Northern region of Ghana. There were 441 (95%) sets of questionnaires completed and received from the students in both Bimbilla Senior High and Wulensi Senior High of an estimated 465 sample size. After examining all the questionnaires, 15 questionnaires lacked substantial information and had to be removed from the analysis and presentation. Therefore, 426 (92%) questionnaires were included in the results and analyzed.

The demographic characteristics included in this study were age, sex/gender, and school. Of the 426 students included in the analysis, 204 (47.9%) were from Bimbilla Senior High and 222 (52.1%) were from Wulensi Senior High. There were 270 (63.4%) male and 156 (36.6%) female participants. The mean age of the students was 18.58 (SD = 1.844) (Range 13–24). The minimum and maximum ages of the 426 students were 13 and 24, respectively.

### Assessment of knowledge of hepatitis B virus

In order for individuals to protect themselves against contracting a disease or condition, and for those who already contracted a disease, to manage the condition effectively, adequate and accurate or comprehensive knowledge of that disease or condition is necessary. We view the knowledge of the rural students about hepatitis B virus infection as a critical step in the prevention of new infections. The results show basic but not a good knowledge of hepatitis B among the participants in this study. Major findings to point out are the responses to four questions that demonstrated a basic rather than good knowledge of HBV. In terms of correct responses, for instance, 295 (69%) answered correctly that hepatitis B is caused by a virus while 131 (30%) participants answered incorrectly by either choosing bacteria or fungus as the cause of hepatitis B infection. For prevention, 344 (80.8%) correctly knew that the most effective way to prevent an HBV infection was by getting vaccinated by a healthcare professional. On the other hand, 228 (53.5%) incorrectly answered that the HBV spread through food. Again, 297 (70%) did not know the number of injections required for being immunized against HBV.

Descriptive analysis showed a mean knowledge score of 11.8 (SD = 1.98). The minimum score was 3 and the maximum score was 16. However, with computing of total knowledge score, 258 (60.6%) scored between 6 and 12 and 161(37.8%) scored between 13 and 16 thereby making up basic knowledge and good knowledge categories respectively. Only 7 (1.6%) demonstrated poor knowledge of HBV. A summary of the categories is presented in Table 2 below.

From the above analysis, it could be seen clearly that most of the participants demonstrated basic rather than a good knowledge of hepatitis B viral infection as described in the definition of knowledge categories above.

We also wanted to know if there was a significant difference in HBV knowledge between male and female students and between students of BSH and WSH, for which we conducted a Mann-Whitney's U test. Male students had a higher mean knowledge rank (215.30) than female students (210.39), however, that difference was not statistically significant (U = 20575, male = 215.30, female = 210.39, *p = 0.688*, two-tailed). Again, a Mann-Whitney's U test showed that the students of WHS had a higher mean knowledge rank (220.18) than the students of BHS (206.23), however, the difference was not statistically significant (U = 21161,

BHS = 206.23, WHS = 220.18, *p = 0.24*, two-tailed). A Pearson correlation showed no relationship between age and HBV knowledge, $r(424) = 0.034$, $p = 0.486$.

## Assessment of risk factors for hepatitis B virus infection

The results of the assessment of risk factors for HBV are presented in Table 3 below. As indicated in the methods section, the risk factors examined included age of the participants, STI diagnosis, number of sexual partners, taking HBV test, engaging in unprotected sexual behavior, unsafe narcotic drug use, and whether or not the participant was vaccinated against HBV at the time of the study. Out of the 426 participants, only three (0.7%) were born after 2003 when mandatory vaccination of infants was introduced and were vaccinated against hepatitis B. The rest were born before 2003 and some of whom may have faced the risk of HBV infection.

A major risk factor that was considered in this study was whether or not the participants had ever been diagnosed with a sexually transmitted infection (STI) prior to the time of the study. This question was added because research has shown that the presence of existing STI compromises the immune system and increases the risk of other infections [28]. From Table 3, 62 (14.6%) participants reported that they had been diagnosed with an STI in the last six months. In terms of the current risky sexual behavior, participants were asked about the number of sexual partners they had and whether or not they had engaged in unprotected sexual behavior in the last six months before the study. The results show that the majority of the students in this study 316 (74.2%) reported they had no sexual partners, and therefore, it was assumed they were not sexually active at the time of the study. However, 102 (23.9%) students reported having one sexual partner at the time of the study while only 8 (1.9%) reported having two or more sexual partners. Of the respondents who were sexually active, 86 (84.3%) reported engaging in unprotected sex with their partner, and therefore increasing their risk of not just hepatitis B infection but other diseases, such as HIV, as well.

Table 3. Risk factors for hepatitis B infection.

| Variable | Frequency | Relative percent |
|---|---|---|
| 1. Have you ever been diagnosed with a sexually transmitted disease? | | |
| Yes | 62 | 14.6 |
| No | 364 | 85.4 |
| 2. How many sexual partners do you currently have? | | |
| None | 316 | 74.2 |
| One | 102 | 23.9 |
| Two or more | 8 | 1.9 |
| 3. Do you inject illegal drugs or have you ever injected illegal drugs? | | |
| Yes | 99 | 23.2 |
| No | 327 | 76.8 |
| 4. Have you ever been tested for Hepatitis B? | | |
| Yes | 86 | 20.2 |
| No | 340 | 79.8 |
| 5. In the last 6 months, have you engaged in unprotected sexual behavior? | | |
| Yes | 86 | 20.2 |
| No | 340 | 79.8 |
| 6. Have you vaccinated yourself against Hepatitis B? | | |
| Yes | 96 | 22.5 |
| No | 330 | 77.5 |

The question of narcotic/illegal drug injection was asked in this study and the results are also presented in Table 3. Out of the 426 participants, 99 (23.2%) students reported that they had injected drugs, elevating their risk of hepatitis B infection. An overwhelming majority (76.8%), however, reported never injecting drugs, and therefore, had a lower risk of a hepatitis B infection compared to those who reported injecting drugs.

The last two questions we included to examine the risk factors of infection were whether they had ever been tested for hepatitis B and whether or not they were vaccinated already against hepatitis B. For testing for the hepatitis B virus, only 86 (20.2%) reported that they had been tested and so knew their status at the time they were tested. Out of the 86 students who reported being screened for HBV, 39 (45.3%) stated they tested positive and were HBV patients. This was a 9.2% prevalence rate among the sample of 426 in this study. Again, the majority of 340 (79.8%) had never tested for HBV and so did not know their status at the time of this study. A similar trend was found with vaccination against HBV whereby only 96 (22.5%) reported they were vaccinated, and therefore, protected. 330 (77.5%) reported they were not vaccinated, and therefore, still faced the risk of infection. From the above discussion, it is clear that the large majority of the rural students in this study were not protected and face the risk of hepatitis B virus infection.

## Vaccination status of participants

One of the goals of this study was to determine the vaccination status of rural students against HBV. The key condition to determine the vaccination status was whether or not they ever received or at least recall being vaccinated against the hepatitis B virus. From the results, an majority of the participants reported not being vaccinated against hepatitis B at the time this study was conducted. Out of the 426 students whose questionnaires were analyzed, only 86 (21%) reported that they tested for HBV. However, in total, 96(22.5%) participants reported being vaccinated against hepatitis B virus. A Pearson $X^2$ showed no association between the knowledge of HBV and taking an HBV test among the participants ($p = 0.928$). There was, however, an association between testing for HBV and vaccinating against HBV ($p < 0.05$).

The model showed that previously tested for the hepatitis B virus ($p < 0.05$) and the high school the student attended (p = 0.032) significantly predicted vaccination against the hepatitis B virus infection. The results are presented in Table 4 below. The variables presented in the table are age (q2), sex/gender(q3), high school attended (q4), ever tested for HBV (q31), number of sexual partners (q35), and total HBV knowledge score(TKSCORE).

## Discussion

The results presented from the study above show a basic but not adequate or comprehensive knowledge of the hepatitis B virus, its modes of infection, and prevention of hepatitis B. The finding of basic rather than good knowledge of HBV in this study is consistent with similar findings in Ghana and other sub-Saharan African countries. For instance, among 175 health-care workers at a public hospital at Suntreso Government Hospital in the Ashanti region of Ghana, knowledge about HBV was reportedly basic and there was lack of adequate knowledge of HBV even among the healthcare workers [22]. Again, among 200 barbers in Obuasi in the Ashanti region of Ghana, 90.5% lacked knowledge about HBV [23]. In Nigeria, a similar study among 643 pregnant women found that 76% reported poor knowledge of HBV. Studies in other parts of the world have reported a similar trend of poor knowledge [16, 17]. Thus, poor knowledge of HBV seems universal. In this study, demographic variables including high school attended, age, and sex had no correlation with knowledge of HBV. Students of WHS and males scored higher on HBV knowledge on the Mann Whitney U test as compared to

**Table 4. Results of logistic regression predicting vaccination against HBV.**

| | | Variables in the Equation | | | | | |
|---|---|---|---|---|---|---|---|
| | | B | S.E. | Wald | df | Sig. | Exp(B) |
| Step 1[a] | q2 | -.056 | .066 | .718 | 1 | .397 | .946 |
| | q3 | -.096 | .261 | .134 | 1 | .714 | .909 |
| | q4 | .533 | .248 | 4.610 | 1 | .032 | 1.704 |
| | q31 | 1.273 | .267 | 22.683 | 1 | .000 | 3.570 |
| | q35 | -.225 | .243 | .857 | 1 | .355 | .798 |
| | TKSCORE | .104 | .060 | 2.981 | 1 | .084 | 1.109 |
| | Constant | -1.530 | 1.546 | .979 | 1 | .322 | .217 |

a. Variable(s) entered on step 1: q2, q3, q4, q31, q35, TKSCORE.

students of BHS and females, respectively, but the differences in knowledge were not statistically significant.

The findings show that a majority of the students were not sexually active at the time of the study, and therefore, most did not engage in risky sexual behavior. However, the risk of infection of hepatitis B in this population was still high because the majority had not been tested and vaccinated against HBV. This is because out of the 426 students, only 86 (20%) reported that they ever screened for HBV while 80% never screened, and most did not know their status with regards to HBV. This finding is also similar to the pattern revealed with the existing literature as many studies have reported low screening. For example, among 9,014 adults sampled from the general population in France, it was reported that only 27% ever screened for HBV [17]. In the study among pregnant women cited earlier, only 20% reported ever screening for HBV [16].

Another important finding in this study was the high prevalence of HBV self-reported by the participants. As presented above, 9.2% of the sample in this study reported being HBV positive and were receiving treatment. This is lower than the 12% national prevalence rate for Ghana reported and many other rates reported across Ghana and other countries. For instance, 10.8% and 11.6% prevalence was reported among 576 voluntary blood donors and blood replacement donors, respectively in Tamale, the Northern regional capital in Ghana [5]. In the Eastern region, 10.6% prevalence was reported among 1500 pregnant women [6]. It was also lower than other prevalence rates reported in Ghana [15, 23, 24].

The ultimate goal of the fight against HBV is to prevent new infections and the most effective means of achieving zero infections is vaccination against the virus. However, the finding in this study is that there is low vaccination among the population and this may be a reflection of the true situation in many rural districts in Ghana. From the results above, only 96 (22.5%) of the 426 students reported being vaccinated. It is, therefore, not out of place to state that the students were largely not protected against hepatitis B infection and that the majority of the students did not know their status with regards to HBV.

The implication of this finding for Ghana is that many people may still be unvaccinated and unprotected and continue to face the risk of HBV infection and meeting sustainable development goal 3 in terms of HBV infection may be elude Ghana. The low vaccination in this study population may be due to the fact that 423 (99.3%) of the participants in this study were born before the mandatory HBV vaccination for all infants was introduced in Ghana in 2003. The age of the participants is unique and important in this study because of the current national policy in Ghana pertaining to immunization against the hepatitis virus. In 2003, Ghana Health Service (GHS) introduced a policy to vaccinate all newborn babies against hepatitis. Prior to 2003, there was no mandatory immunization against hepatitis B and so all those

participants who were born before 2003 stood the risk of being infected unless they voluntarily vaccinated themselves against it.

Finally, testing for HBV and attending BHS were found to be significantly associated with being vaccinated against the virus. This underscores the importance of HBV screening programs as a critical strategy to motivating vaccination and reducing infections. Students of BHS were more likely to be vaccinated against HBV than students of WHS. This may be due to exposure to programs about HBV and district hospital in Nanumba North District as compared to Nanumba South District.

The results in this study are significant in helping to highlight the gaps in the sustainable development goal (SDG) 3.3.4 in terms of reducing infectious diseases including HBV infection. Indeed significant progress was reported in terms of global HBV incidence and prevalence whereby newborn infections of chromic HBV reduced from 4.7% during the pre-vaccination era to 0.8% in 2017 [6]. However, there is a gap that needs to be bridged. There are still so many people who were born prior to the mass vaccination initiation and who continue to face the risk of infection as found in this study.

## Limitations

A major limitation we encountered was the location for the data collection. Students sat in classrooms usually arranged for examinations to complete the questionnaire and this may have impacted their responses. Completing the questionnaire may have seemed like writing an official exam and so they may have felt pressured to complete the questionnaires. Besides, since pretesting of the questionnaire was done with a few students from the schools, we felt that they may have discussed it with some of the peers who ended up being recruited into the study. These notwithstanding, the data collection went through without issues as the participants completed the questionnaires independently and freely. We had explained to the students that participation was completely voluntary as indicated above and that it was not an examination in any form.

## Conclusion

While we exercise caution in generalizing the results of this study to all rural high school students in Ghana, the results of this study have re-echoed the high prevalence of HBV in Ghana. The poor state of knowledge and a high risk of HBV infection among young adults in rural communities have also been highlighted in the findings of this study. The results have also shown that vaccination against HBV infection is low among rural students and young adults at least in the Nanumba North and South districts of Ghana. Finally, HBV screening is shown to be significantly associated with vaccination against the virus. There is a need for culturally appropriate and evidence-based educational interventions to improve the knowledge of HBV. Developing and implementing a national HBV screening and vaccination programs are critical in winning the fight against the increasing morbidity and mortality caused by HBV infections in Ghana.

## Acknowledgments

We wish to sincerely thank all those who supported us in the organization and collection of data for this study. We particularly wish to thank Miss Doris Hadzi and Mr. Kingsley Afeti of School of Public Health of the University of Health and Allied Sciences, who helped in questionnaire printing and data entry.

## Author Contributions

**Conceptualization:** Awolu Adam.

**Data curation:** Awolu Adam.

**Formal analysis:** Awolu Adam.

**Investigation:** Awolu Adam, Adam Fusheini.

**Methodology:** Awolu Adam.

**Project administration:** Awolu Adam.

**Resources:** Awolu Adam, Adam Fusheini.

**Supervision:** Adam Fusheini.

**Validation:** Awolu Adam, Adam Fusheini.

**Writing – original draft:** Awolu Adam.

**Writing – review & editing:** Adam Fusheini.

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
