## [Decision Letter · Decision Letter 0]

31 Oct 2019

PONE-D-19-25602

Knowledge, Risk of Infection, and Vaccination Status of Hepatitis B Virus among Rural High School Students in the Nanumba North and Nanumba South Districts of Ghana

PLOS ONE

Dear Dr Adam,

Thank you for submitting your manuscript to PLOS ONE. After careful consideration, we feel that it has merit but does not fully meet PLOS ONE’s publication criteria as it currently stands. Therefore, we invite you to submit a revised version of the manuscript that addresses the points raised during the review process. Some important points needs to be clarified.

We would appreciate receiving your revised manuscript by 2 months. To enhance the reproducibility of your results, we recommend that if applicable you deposit your laboratory protocols in protocols.io, where a protocol can be assigned its own identifier (DOI) such that it can be cited independently in the future. For instructions see: http://journals.plos.org/plosone/s/submission-guidelines#loc-laboratory-protocols

We look forward to receiving your revised manuscript.

Kind regards,

Isabelle Chemin, PhD

Academic Editor

PLOS ONE

Journal Requirements:

2. Please include further details concerning the development or pre-testing of your questionnaire. For example, upon whom was this performed and on how many participants.

3. Please expand on why a sample size of 426 was used in this study - e.g. a power calculation was performed prior to participant recruitment. Additionally, please refer to any post-hoc corrections made during your statistical analysis. Please justify the reasons if these were not performed, and refrain from reporting p-values as 0.000. Either report the exact value or use the format p<0.001.

4. Please make sure you have fully explored any limitations of the study within your Discussion.

5. You indicated that you had ethical approval for your study. In your Methods section, please ensure you have also stated whether you obtained consent from parents or guardians of the minors included in the study. Please ensure you have specified (1) whether consent was informed and (2) what type you obtained (for instance written or verbal, and if verbal how this was documented and witnessed).

6. We suggest you thoroughly copyedit your manuscript for language usage, spelling, and grammar. If you do not know anyone who can help you do this, you may wish to consider employing a professional scientific editing service.  

7. We note that you have indicated that data from this study are available upon request. PLOS only allows data to be available upon request if there are legal or ethical restrictions on sharing data publicly. For information on unacceptable data access restrictions, please see http://journals.plos.org/plosone/s/data-availability#loc-unacceptable-data-access-restrictions.

8. Please amend either the abstract on the online submission form (via Edit Submission) or the abstract in the manuscript so that they are identical.

Additional Editor Comments (if provided):

Reviewers' comments:

Reviewer's Responses to Questions

**Comments to the Author**

1. Is the manuscript technically sound, and do the data support the conclusions?

Reviewer #1: Partly

2. Has the statistical analysis been performed appropriately and rigorously? 

Reviewer #1: No

3. Have the authors made all data underlying the findings in their manuscript fully available?

Reviewer #1: No

4. Is the manuscript presented in an intelligible fashion and written in standard English?

Reviewer #1: No

5. Review Comments to the Author

Reviewer #1: Short Review Report

The paper seeks to assess the knowledge, vaccination status and risk factors of HBV amongst high school students in a bid to identify knowledge gaps and influence policy of HBV vaccination. The study is important considering the fact the effective interventions targeted at raising awareness towards HBV are predicated on first identifying knowledge gaps and practices. The authors efforts are commendable. 

Please refer to the comments in the edited manuscript for major comments 

General Comments

Title: The title of the manuscript properly captures the scope of the study. 

Introduction: The authors in their introduction try to make a case for the public health importance of HBV in Sub-Saharan Africa highlighting the dearth of information in the study region needed to influence policy and make necessary changes. While this is the aim of the authors in the introduction, it might be a bit difficult for readers to immediately see this as the introduction in poorly written with grammatical errors and lack of cohesiveness between sentences and paragraphs. The authors should take some time to revise the entire manuscript for clarity. 

Methods: The sampling strategy used by authors in this study is well founded and appropriate for this type of study. The information in the methods section is however lacking in many critical details some of which the authors decided to place in other sections of the manuscript. The authors should for example, explain how the knowledge score is computed. The authors should consolidate the methods scattered all over the manuscript into the methodology section ensuring that the flow of information is logical and easy to follow. Finally, authors should ensure due diligence is performed in selecting the appropriate statistical tests. For example, authors should provide evidence that dependent variable are normally distributed before using parametric tests such as t test. 

Results: The authors have some interesting results however, authors should work at presenting the results better. Authors should refrain from repeating results already presented in tables in the main text. Some critical results such as the regression table are missing from the manuscript. 

Discussion: Authors have tried to relate this study to previously published studies however authors need to also discuss the implications of their findings for their local setting, country and maybe the world at large. The authors might also want to consider including limitations of their study.

6. PLOS authors have the option to publish the peer review history of their article (what does this mean?). If published, this will include your full peer review and any attached files.

Reviewer #1: No

---

## [Author Response · Author response to Decision Letter 0]

28 Dec 2019

The response to reviewers comments have been attached here.

---

## [Decision Letter · Decision Letter 1]

7 Feb 2020

PONE-D-19-25602R1

Knowledge, Risk of Infection, and Vaccination Status of Hepatitis B Virus among Rural High School Students in the Nanumba North and Nanumba South Districts of Ghana

PLOS ONE

Dear Dr Adam,

Thank you for submitting your manuscript to PLOS ONE. After careful consideration, we feel that it has merit but does not fully meet PLOS ONE’s publication criteria as it currently stands. Therefore, we invite you to submit a revised version of the manuscript that addresses the minor points raised during the review process.

We would appreciate receiving your revised manuscript by 1 month. To enhance the reproducibility of your results, we recommend that if applicable you deposit your laboratory protocols in protocols.io, where a protocol can be assigned its own identifier (DOI) such that it can be cited independently in the future. For instructions see: http://journals.plos.org/plosone/s/submission-guidelines#loc-laboratory-protocols

We look forward to receiving your revised manuscript.

Kind regards,

Isabelle Chemin, PhD

Academic Editor

PLOS ONE

Reviewers' comments:

Reviewer's Responses to Questions

**Comments to the Author**

1. If the authors have adequately addressed your comments raised in a previous round of review and you feel that this manuscript is now acceptable for publication, you may indicate that here to bypass the “Comments to the Author” section, enter your conflict of interest statement in the “Confidential to Editor” section, and submit your "Accept" recommendation.

Reviewer #1: (No Response)

2. Is the manuscript technically sound, and do the data support the conclusions?

Reviewer #1: Yes

3. Has the statistical analysis been performed appropriately and rigorously? 

Reviewer #1: Yes

4. Have the authors made all data underlying the findings in their manuscript fully available?

Reviewer #1: Yes

5. Is the manuscript presented in an intelligible fashion and written in standard English?

Reviewer #1: Yes

6. Review Comments to the Author

Reviewer #1: I want to commend the authors for taking the time to revise this manuscript. Authors have addressed most of the comments raised in the previous review. A few extra comments have been made in the manuscript.

7. PLOS authors have the option to publish the peer review history of their article (what does this mean?). If published, this will include your full peer review and any attached files.

Reviewer #1: No

---

## [Author Response · Author response to Decision Letter 1]

24 Feb 2020

We have attached our response to reviewers as a word document.

---

## [Editor Report · Decision Letter 2]

6 Apr 2020

Knowledge, Risk of Infection, and Vaccination Status of Hepatitis B Virus among Rural High School Students in the Nanumba North and Nanumba South Districts of Ghana

PONE-D-19-25602R2

Dear Dr. Awolu,

We are pleased to inform you that your manuscript has been judged scientifically suitable for publication and will be formally accepted for publication once it complies with all outstanding technical requirements.

With kind regards,

Prof. Maria Gańczak

Academic Editor

PLOS ONE
---

## [Editor Report · Acceptance letter]

10 Apr 2020

PONE-D-19-25602R2 

Knowledge, risk of infection, and vaccination status of hepatitis B virus among rural high school students in Nanumba North and South Districts of Ghana 

Dear Dr. ADAM:

I am pleased to inform you that your manuscript has been deemed suitable for publication in PLOS ONE. Congratulations! Your manuscript is now with our production department. 

With kind regards,

on behalf of

Prof. Maria Gańczak 

Academic Editor

PLOS ONE